# Nucleosome dynamics of human iPSC during neural differentiation

Janet C Harwood[1] 🆔, Nicholas A Kent[2] 🆔, Nicholas D Allen[2] 🆔 & Adrian J Harwood[2,3,*] 🆔

## Abstract

Nucleosome positioning is important for neurodevelopment, and genes mediating chromatin remodelling are strongly associated with human neurodevelopmental disorders. To investigate changes in nucleosome positioning during neural differentiation, we generate genome-wide nucleosome maps from an undifferentiated human-induced pluripotent stem cell (hiPSC) line and after its differentiation to the neural progenitor cell (NPC) stage. We find that nearly 3% of nucleosomes are highly positioned in NPC, but significantly, there are eightfold fewer positioned nucleosomes in pluripotent cells, indicating increased positioning during cell differentiation. Positioned nucleosomes do not strongly correlate with active chromatin marks or gene transcription. Unexpectedly, we find a small population of nucleosomes that occupy similar positions in pluripotent and neural progenitor cells and are found at binding sites of the key gene regulators NRSF/REST and CTCF. Remarkably, the presence of these nucleosomes appears to be independent of the associated regulatory complexes. Together, these results present a scenario in human cells, where positioned nucleosomes are sparse and dynamic, but may act to alter gene expression at a distance via the structural conformation at sites of chromatin regulation.

**Keywords** chromatin; human iPSC; neural differentiation; nucleosome positioning

**Subject Categories** Chromatin, Epigenetics, Genomics & Functional Genomics; Development & Differentiation; Stem Cells

## Introduction

Placement of nucleosomes has been implicated in gene regulation from *Saccharomyces cerevisiae* [1] to human cells [2], and mutations in chromatin remodelling enzymes that control nucleosome positioning have a demonstrable role in gene regulation. Chromatin remodelers of the SNF2 protein family have strong associations with human disease, including mental health and cancer biology [3,4].

The SNF2 protein Brg-1 is associated with a range of psychiatric disorders and intellectual disabilities (ID), such as Coffin–Siris syndrome [5]. More strikingly, the chromodomain-helicase-DNA binding (CHD) sub-family proteins are intimately associated with neuropsychiatric or neurodevelopmental disorders [3]. CHD2 and CHD4 increase the risk of epileptic encephalopathies, and CHD2, along with CHD5 and CHD6, confer risk for ID. CHD7 is causative of CHARGE syndrome [6,7], a multi-system, developmental disorder, but also imparts genetic risk for autism spectrum disorder (ASD) as does CHD1, CHD2 and CHD3. Finally, CHD8 has a strong association with both ASD and schizophrenia [8,9]. Ultimately to understand why aberrant chromatin remodelling is important for disease risk, we need to establish the underlying principles that determine the relationship between nucleosome positioning and gene regulation.

Nucleosome interactions within chromatin are likely to be complex but can be described by the following quantitative parameters: *position*, the sequence coordinates where nucleosomes occur across the cell population; *occupancy*, the frequency that a nucleosome is present at an individual site; and *accessibility*, the degree to which the presence of nucleosomes restricts access to the DNA. In organisms with small genomes, such as in the yeasts and *Dictyostelium*, nearly 80% of nucleosomes are positioned in the same place in the genome across the cell population, forming arrays that span gene bodies [10–13]. There are distinct regions of low occupancy and increased accessibility, for example in the nucleosome-free region (NFR) found at the transcriptional start site (TSS) of genes. Finally, there is evidence for mutual exclusivity between a positioned nucleosome and transcription factor (TF) binding at the same site [14]. Nucleosomes at active genes appear to be more positionally organised than at inactive genes and changes in nucleosome positioning, such as the specific increase in nucleosome spacing seen in a *Dictyostelium* mutant lacking the CHD8 homologue, ChdC, alter gene expression. Nonetheless, even in these organisms the actual correlation between positioning and gene expression can be low; for example, only 15% of genes with altered nucleosome spacing in a *chdC* mutant show corresponding changes in gene expression [12].

In mammalian cells, including human cells, with large and complex genomes, non-coding and intragenic DNA sequences can make up approximately 98% of the genome. In these larger and more complex genomes, the pattern of positioned nucleosomes

1 MRC Centre for Neuropsychiatric Genetics & Genomics, Cardiff University, Cardiff, UK
2 School of Biosciences, Cardiff University, Cardiff, UK
3 Neuroscience and Mental Health Research Institute (NMHRI), Cardiff University, Cardiff, UK
  *Corresponding author. Tel: +44 2920688492; E-mail: harwoodaj@cf.ac.uk

appears to be radically different. Studies in human cells have shown that only a very small proportion of the total nucleosome number are strongly positioned [15,16]. Although the nucleosome patterns flanking TSSs can conform to those of non-metazoan organisms, arrays within gene bodies have not been widely reported. Studies also suggest that occupancy and accessibility may be uncoupled. Mapping DNA accessibility in chromatin by its sensitivity to DNase digestion can reveal hypersensitive sites, which are often found at gene regulatory elements distant from the gene body. Other regions, although not hypersensitive, show increased sensitivity to MNase digestion; hence, accessibility of proteins is higher in these regions of DNA. These MNase-sensitive regions coincide with regions of open chromatin as defined by ATAC-seq, certain chromatin marks and transcriptional activity [17]. Interestingly, such MNase accessibility maps (MACC) show that increased accessibility can occur without changes of nucleosome occupancy [18].

Here, we focus on the relationship between gene expression and nucleosome positioning, rather than accessibility, in the context of human neural differentiation. Unlike the relationship of MACC to open chromatin and gene expression, the relationship of these positioned nucleosomes to gene expression is unclear. By comparing maps of well-positioned nucleosomes with low MNase sensitivity for the same human iPSC line in the pluripotent state and following differentiation to neural progenitor cells (NPCs), we have directly examined the developmental dynamics of nucleosome positioning during early neural differentiation and its relationship to gene expression.

## Results and Discussion

### Positional mapping of nucleosomes during human neural differentiation

To generate nucleosome maps based on positioning, we used a modified MNase-seq methodology, where chromatin is rapidly digested with MNase *in situ* in permeabilised cells without cross-linking [11]. Bulk chromatin of pluripotent iPSC, and following their differentiation to NPCs, was digested with MNase to an equivalent degree, giving near identical levels of mono-nucleosome fragments (Fig EV1A). Size-fractionated, MNase-protected DNA digestion fragments were paired-end-sequenced to generate between 2.25 and $2.5 \times 10^9$ paired-end reads per digest, and the positions of the fragment mid-points were mapped to the genome. To monitor the initial digestions, the abundance of fragment sizes was plotted for both cell types (Fig EV1B), and directly compared in the size classes that span the range of fragment sizes seen for human nucleosomes (Fig EV1C). This confirmed comparable degrees of digestion between the two cell states in the mono-nucleosome size range, but with a slight bias towards larger fragments in the pluripotent cell samples.

To map nucleosome positions in the genome, we plotted the sequence read mid-point frequency distribution of fragments in the 138–161 bp range, corresponding to the nucleosome footprint size (Fig 1A). We developed a heuristic, peak-finding algorithm (PeakFinder) to identify highly positioned nucleosomes, based on peak shape and read depth. This identified more than 400,000 highly positioned nucleosomes in NPCs (Fig 1B). For brevity, we

will refer to these nucleosomes as "positioned". Using this tool on published MNase-seq datasets from the human cell lines, K562 and GM12878 [19], we identified broadly similar numbers of positioned nucleosomes to those reported in their associated publication (Table EV1). Thus, we estimate that 2.7, 2.4 and 1.6% of nucleosomes are positioned in human NPC, K562 and GM12878 cells, respectively. Based on a theoretical total of 15 million nucleosomes in the human genome, assuming 1 nucleosome per 200 bp. This is consistent with observations from other human cells [15,16].

In contrast, there were 8.4-fold fewer positioned nucleosomes (0.33% of the total nucleosome number) in the pluripotent cell state, indicating that a substantial increase in positioned nucleosomes occurs during differentiation from pluripotent cells to NPC. We recognise that it may be difficult to detect positioned nucleosomes that have high MNase sensitivity using our methodology. To investigate the impact of this possibility on the detected nucleosome number, we mapped the mid-points of DNA fragments in the 112–137 bp sub-nucleosome size range (Fig 1B). We found no evidence for increased cleavage of nucleosomes or accumulation of sub-nucleosome size fragments in pluripotent iPSC. Furthermore, in the total population of DNA fragments, there was no accumulation of small DNA fragments (Fig EV1B).

Comparison of both pluripotent iPSC and NPC maps indicated that only 32% of the positioned nucleosomes present in the pluripotent state retain their position during differentiation to NPC (Fig 1C). This indicates that there are considerable changes in nucleosome positioning during human neural differentiation due to both re-positioning and the *de novo* formation of positioned nucleosomes. Importantly, there is a small population of nucleosomes that retain their positions in both cell states, albeit corresponding to only approximately 0.1% of the total nucleosome population.

### Global organisation of nucleosome positioning

To determine how human nucleosomes are distributed throughout the genome, the organisation of positioned nucleosomes was analysed in pluripotent iPSC and NPC states, and for the K562 cell line. A frequency distribution was plotted for all nucleosome positions and their surrounding window ($\pm$ 300 bp) centred on the nucleosome peak (Fig 1D). This showed that very few nucleosomes appear in evenly spaced arrays. To quantify the number and length of positioned nucleosome arrays, we calculated the frequency that positioned nucleosomes occur with a spacing of 50 bp or less (Fig 1E) and the distribution of their inter-nucleosome spacing from 0 to 100 kb (Fig EV2). Approximately 90% of positioned nucleosomes are present as singletons, and few nucleosomes (7.1, 7.0 and 13.1%, respectively) occur in pairs or nucleosome arrays. The same basic pattern of nucleosome positioning is seen across the different human cell types tested here (Fig 1D and E). This distribution contrasts with the arrays of nucleosomes that are seen across gene bodies in organisms with small genomes. Nonetheless, in total there are in fact more positioned nucleosomes in the human genome than in the yeast genome. This apparent sparsity of positioned human nucleosomes may arise from the need to precisely position nucleosomes at key regulatory sites rather than organise nucleosome arrays on active genes.

To probe the relationship between nucleosome positions and gene activity further, we examined the distribution of positioned

                                                                      

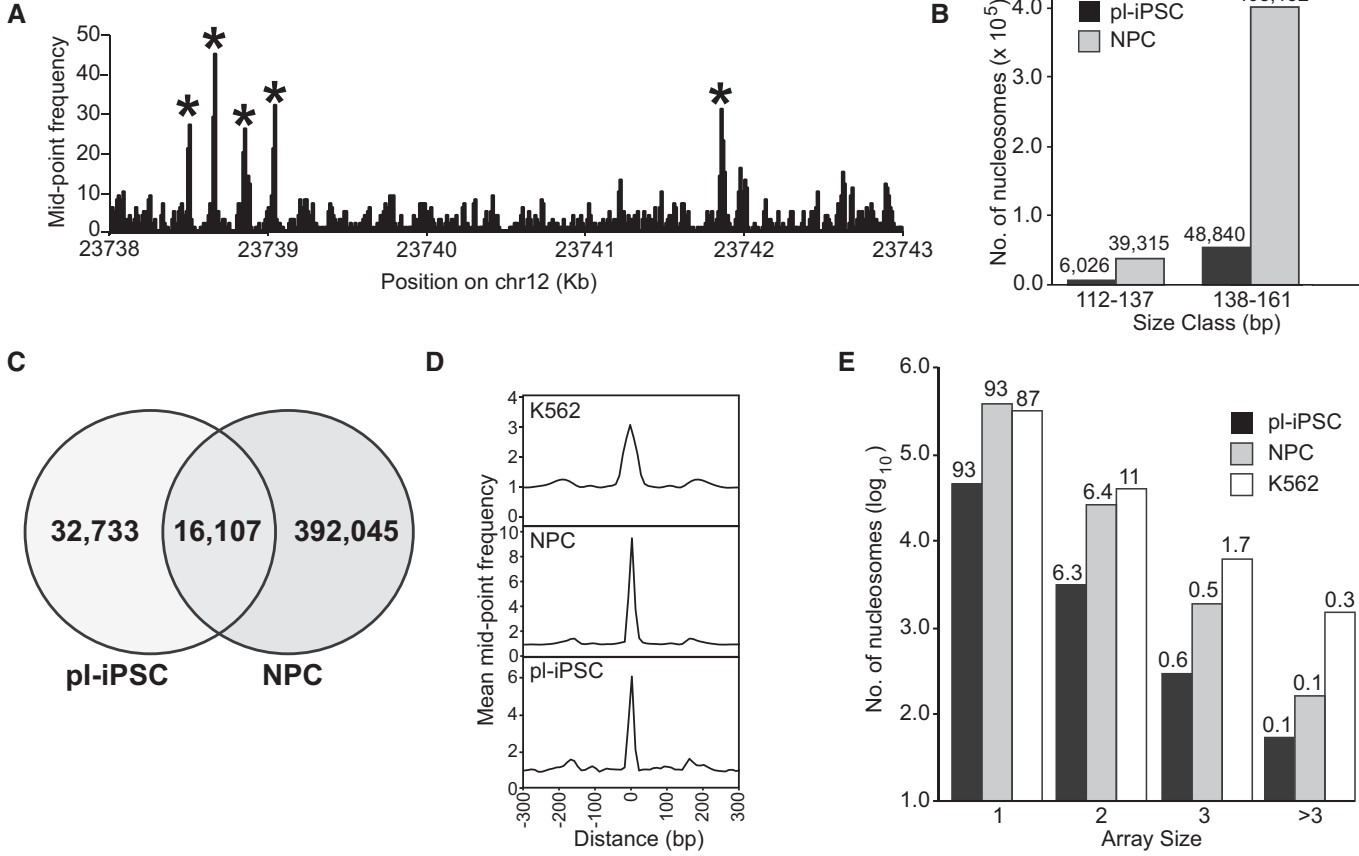

**Figure 1. Nucleosome dynamics of human pluripotent stem cell and differentiated cells.**

A   Genome-wide nucleosome maps were generated from the frequency distribution of the read mid-point positions of paired-end sequenced MNase-resistant fragments in the size range 138–161 bp (spanning the nucleosome footprint). Figure shows a section of a nucleosome map taken from chromosome 12 derived from pluripotent iPSC (pl-iPSC) to show examples of nucleosomes selected as highly positioned, marked*.

B   The distribution of positioned nucleosomes (derived 138–161 bp size class) and sub-nucleosome fragments (derived 112–137 bp size class) for pl-iPSC- and iPSC-derived NPC (NPC). Actual calculated numbers are presented above each column.

C   Venn diagram showing the overlap in the genomic location of positioned nucleosomes in pl-iPSC and in the same cell line differentiated into neural progenitor cells (NPCs).

D   Average frequency distribution of nucleosome positions relative to each other. Data were aligned to each mapped nucleosome in the genome and plotted for a 600-bp window for pl-iPSC, NPC and the chronic myelogenous leukaemia (Cml)-derived cell line, K562. In all cases, a prominent single nucleosome is presented with only minor flanking peaks, indicating that the majority of nucleosomes do not occur as evenly spaced arrays in human cells.

E   Distribution of nucleosome array sizes for pl-iPSC, NPC and K562, calculated as the number of nucleosomes within a distance of 150-200 bp of each other. Bars show the number of nucleosomes arrayed as singletons (1), pairs (2), triplets (3) and greater than 3 (> 3), displayed as $\log_{10}$ values, and the percentage distribution within a cell line is shown above the column.

nucleosomes across different chromatin states. Using the 15-state model of Ernst and Kellis [20,21], which classifies chromatin from highly active to repressed states based on modifications, functions and associated proteins, we investigated whether positioned nucleosomes partition into particular chromatin states in pluripotent cells. We found no significant enrichment for the major chromatin states associated with transcriptionally active, repressed or heterochromatin. An exception was for chromatin state 8, where we observed an approximate sixfold enrichment of nucleosome positioning over that expected for random placement in the genome (Fig 2A). This state is characterised by its association with the chromatin architectural protein CCCTC-binding factor (CTCF) [22] (Fig 2B). Analysis of our NPC data showed that despite an eightfold increase in the number of

positioned nucleosomes, the increase was distributed evenly across all states, with no particular state showing a strong proportional increase (Table EV2).

Finally, we examined the relationship between nucleosome positioning and open and closed chromatin. Using the map coordinates for open chromatin regions (based on high ATAC-seq accessibility) for pluripotent human embryonic stem cells (H9), [23] we calculated that 5.7% of positioned nucleosomes fall within regions of open chromatin, but given that only 0.72% of chromatin is in the open state, this is an eightfold enrichment over that expected by random placement (Fig 2A). 40% of CTCF sites are in the open state (Fig 2B), and the overlap between positioned nucleosomes with CTCF sites could substantially contribute to their partitioning into open chromatin.

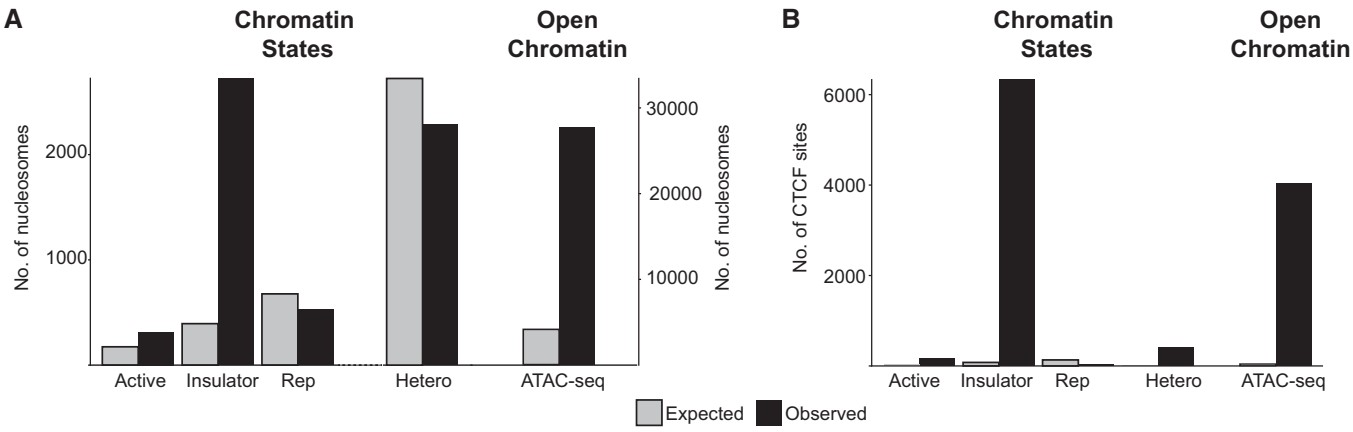

**Figure 2. Positioned nucleosomes, chromatin states and transcriptional activity.**

A   Bar chart showing the number of nucleosomes that map to selected chromatin states [20]. Active (state 1); Insulator (state 8); Repressed, Rep (state 12) and heterochromatin, Hetero (state 13), and to open chromatin (based on ATAC-seq). Shown are the observed number of nucleosomes per chromatin state (black) and the expected number calculated assuming a random distribution across the genome (grey).

B   Bar chart showing the number of CTCF sites that map to different chromatin states and open chromatin, using the same methodology as (A).

## The relationship between nucleosome positioning and gene activity

The poor correlation between the presence of positioned nucleosomes and transcriptionally active chromatin states was unexpected given the known association of nucleosome positioning within active gene loci of non-mammalian cells and the association of gene expression with SNF-2 family chromatin remodelers [24]; therefore, we investigated the local distribution of nucleosome positioning at TSS. In non-mammalian cells, actively transcribed genes possess a positioned nucleosome at their TSS, termed the +1 nucleosome. In the human genome, we identified individual loci where positioned nucleosomes were present flanking the TSS. For example, a nucleosome was present at the −1 position in the pluripotent cell state but not in NPC at the TSS of the pluripotent-specific gene NANOG. (Fig 3A). Conversely, we found individual loci where a positioned nucleosome was present at the +1 position only in NPC, such as at the TSS of the ionotropic glutamate receptor subunit GRIA1 (Fig 3B). However, this correlation did not hold at the whole genome level.

We conducted a global analysis of chromatin patterns at the TSS of protein-coding genes, using a non-redundant list of 66,047 mapped human TSS. The frequency distribution of MNase-protected fragments at and surrounding protein-coding TSS showed a distinct MNase-hypersensitive region at the TSS (Fig 3C), a NFR, as reported previously by others [25]. We also noted distinct nucleosome peaks flanking the NFR and TSS, sitting at positions −1, +1 and +2. This demonstrates the presence of distinctive consensus nucleosome pattern at the TSS, to which our two gene examples, NANOG and GRIA1, conform.

This raises the question of whether positioned nucleosomes in general are required for active gene expression. This can be addressed by comparison between our two developmental states by examining genes expressed only in the pluripotent or NPC state. By cross-comparison of published RNA-seq data from human cells [26] with the non-redundant TSS list of all human TSS ($n = 83{,}179$), we created

datasets containing the locations of TSS of human genes that are transcribed only in pluripotent or NPCs. These datasets were then filtered for the presence or absence of a positioned nucleosome within ± 300 bp of the TSS. The results of this analysis show that for genes expressed only in pluripotent cells there were in fact more nucleosomes positioned at the TSS of genes inactive in the NPC state than for the same genes when they are actively expressed in pluripotent state (Figs 3D and EV3). For genes expressed only in NPCs, a substantial number of NPC-specific genes had nucleosomes associated with the gene TSS in the inactive, pluripotent cell state (Figs 3D and EV3).

Overall, we did not see a strong correlation between the presence of positioned nucleosomes at the TSS and gene expression. In contrast to nucleosome modifications [27,28], the presence of highly positioned nucleosomes at TSS is not a genome-wide predictor of gene activity in human cells. Our findings also mirror the conclusions of [18] and co-workers, who found that nucleosome accessibility, not occupancy, is the predominant predictor of gene expression for UPR-associated gene activation in *Drosophila*.

### Nucleosome positioning at NRSF/REST-binding sites

To widen our analysis further, we examined the patterns of nucleosome positions at selected transcription factor (TF) binding sites involved in neural differentiation, namely YY1, ATF2 and PAX6 [29–31]. We did not observe a pattern of highly positioned nucleosomes flanking these TF binding sites (Fig EV4). In contrast, well-positioned nucleosomes were present flanking the RE1 binding site [32] of NRSF/REST (Fig 4). NRSF/REST promotes epigenetic repression of gene activity by binding to DNA and acting as a scaffold for a protein complex containing enzymes mediating repressive nucleosome modifications, including the H3K9 dimethyltransferase G9a/EHMT2, the histone deacetylase complex Sin3/HDAC1/2 and the H3K4 demethylase LSD1 as well as the chromatin remodeller Brg1/SMARCA4 [33].

RE1 sites were selected by mapping the consensus sequence of RE1 sites to NRSF/REST ChIP-seq data [34] creating a dataset of 871

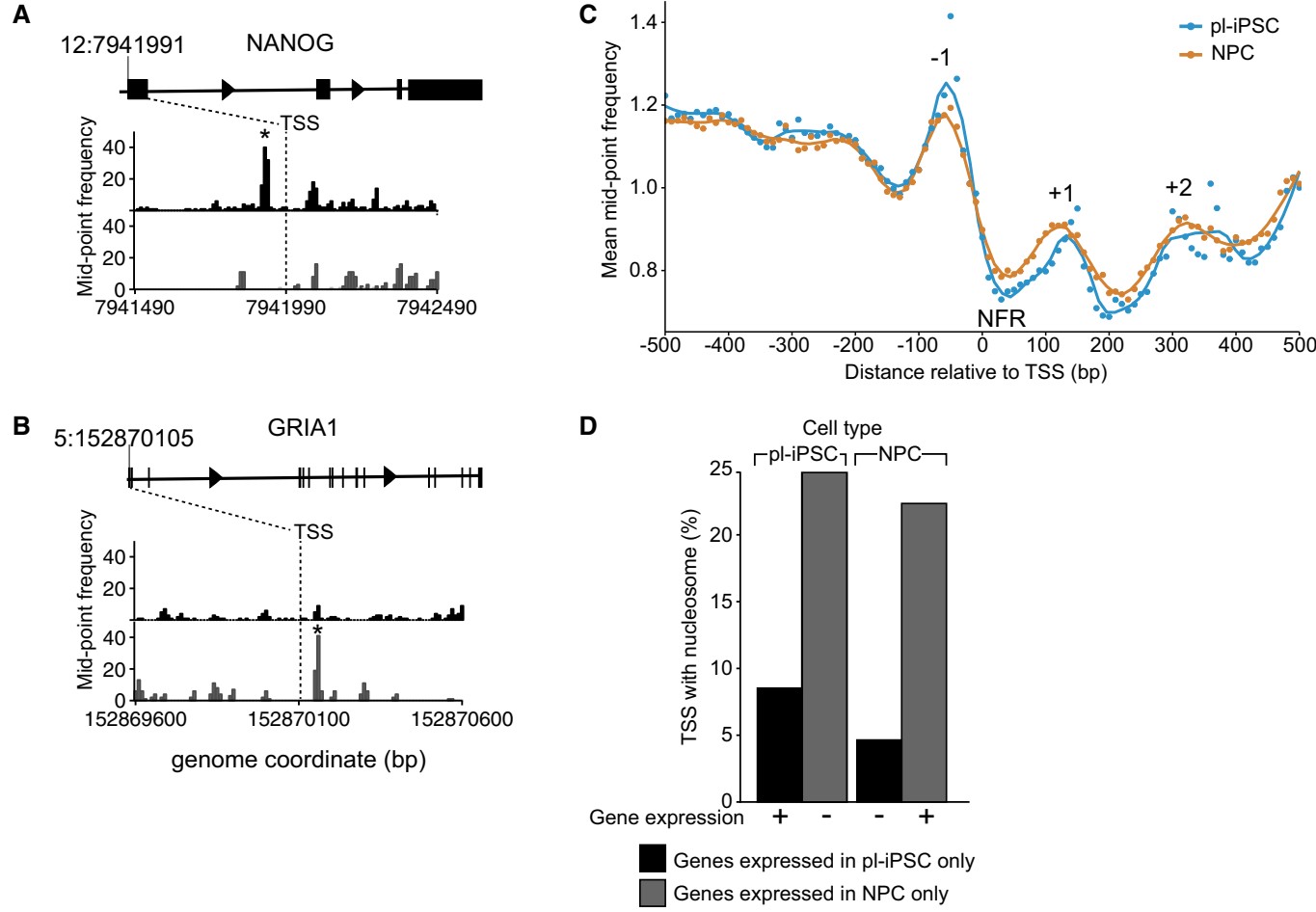

**Figure 3. The relationship of positioned nucleosomes at the TSS and gene expression.**

A, B  Nucleosome maps at and surrounding, the TSS region of the NANOG and GRIA1 genes in pl-iPSC (upper, black) and in NPCs (lower, grey). The TSS is indicated by a dashed line at chr 12: 7,941,991 for NANOG and chr5: 152,870,105 for GRIA1. * indicates the position of a highly positioned nucleosome associated with the TSS of the active gene.

C  Global frequency distribution of nucleosome distributions at TSS of protein-coding genes. Alignment of nucleosome maps centred on TSS using the non-redundant list of 66,047 mapped human protein-coding gene TSS. The features corresponding to the nucleosome-free region (NFR) and −1 to +2 nucleosomes are marked.

D  Genome-wide positioning of nucleosomes within ± 300 bp of a TSS of an active gene. The number of nucleosomes observed is plotted for both pl-iPSC and NPC types for gene sets selected either as uniquely expressed in pl-iPSC (n = 3,833) or NPC (n = 2,082). If positioned nucleosomes are a strong predictor of gene activity, high numbers of nucleosomes would be expected for pluripotent-specific genes in pl-iPSC, but not NPC stages, and for NPC-specific genes in NPC, but not pl-iPSC stages. No correlation between gene activity and positioned nucleosomes was observed.

well-characterised RE1 sites. Aligning nucleosome positioning data to these RE1 sites across the genome revealed a consensus pattern comprising an array of nucleosomes flanking the RE1 site (Fig 4A). The NRSF/REST complex is present at RE1 sites in mouse pluripotent cells [35], but it is lost upon neuronal differentiation. It was therefore unexpected to find that the pattern of nucleosomes surrounding the RE1 site in human pluripotent cells was still present in NPC (Fig 4A). To analyse this pattern in more detail, we stratified sequencing reads into the fragments in the 122–137 bp range for protection by transcription factors or regulatory complexes with large DNA footprints and identified a distinct peak of protected fragments in this smaller size-class at the RE1 site in pluripotent cells corresponding to the NRSF/REST complex. This peak is reduced by more than 90% in NPCs (Fig 4B). To separate the NRSF/REST footprint from that of the

flanking nucleosomes, we constructed a nucleosome map using only larger fragment sizes (162–188 bp), minimising the overlap with the smaller 122- to 137-bp fragments (Fig 4C). This shows a complete separation of NRSF/REST binding from nucleosome positions and significantly that the average pattern of nucleosome positioning is unchanged in NPC in the absence of the NRSF/REST complex.

To ensure that genome averaging did not mask any differences at these sites, we carried out cluster analysis (Fig 4D) to establish that 64% of the selected RE1 sites had detectable peaks, corresponding to NRSF/REST complex in undifferentiated pluripotent cells. These were absent at the same sites at the NPC stage. In pluripotent cells, there was a strong correlation between the presence of NRSF/REST at the RE1 binding site and an array of well-positioned nucleosomes. This nucleosome array pattern was retained at the same sites in

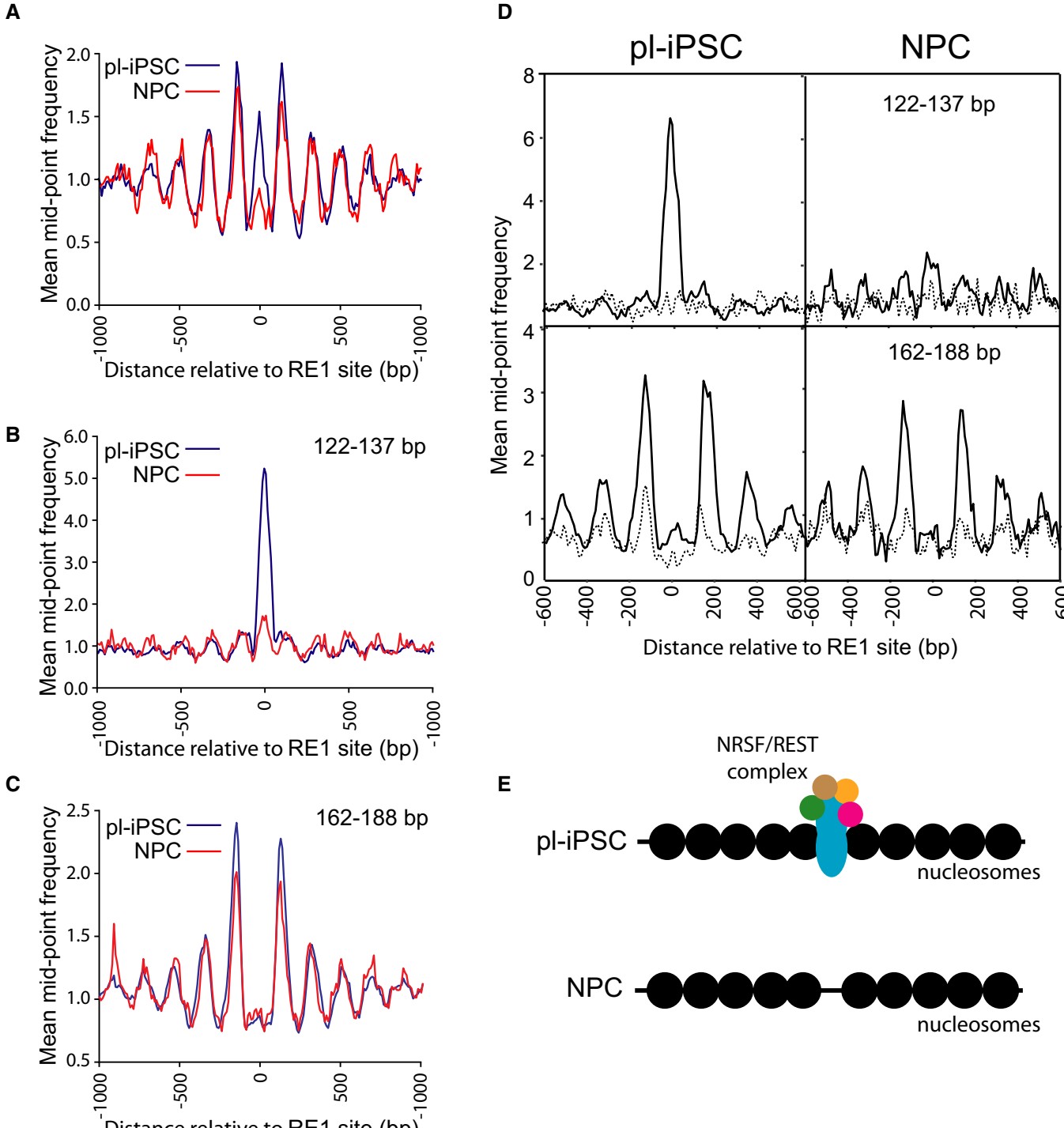

**Figure 4. Nucleosome patterning associated with the RE1 NRSF/REST-binding motif.**

A   The average distribution of nucleosomes centred on the RE1 site (*n* = 871), spanning 2 kb of flanking genomic sequence.

B, C   Aligned paired-end sequence read data were stratified into two size classes, (B) 122–137 bp and (C) 162–188 bp for pl-iPSC and NPC. Plots show the average distribution of nucleosomes centred at and surrounding the RE1 site (*n* = 871). Peaks correspond to the NRSF/REST complex, and nucleosomes in (B) and (C), respectively.

D   Cluster analysis of nucleosome positioning centred on the RE1 site ± 300 bp. RE1 sites were clustered based on the 122–137 bp mid-point sequence read values (NRSF/REST protein complex) from pl-iPSC, shown as a frequency distribution in upper left panel. The frequency distribution data for the same sites were plotted using 122–137 bp data from NPC (upper right panel) and 162–188 bp data (nucleosomes; bottom panels) for the strongest (solid line) and weakest (dotted line) clusters.

E   Schematic of nucleosome patterning at the RE1 site in pl-iPSC and NPC. Nucleosomes are represented by filled black circles.

NPCs, despite the absence of the NRSF/REST complex. These observations indicate that NRSF/REST-binding sites are associated with a distinct consensus nucleosome positioning pattern; however, this pattern is not dependent on the presence of high levels of the NRSF/REST protein complex (Fig 4E). The function of this retained chromatin structure is unknown. However, it is possible that this structure might persist so that the protein complex can rapidly re-engage in the neuronal state, where it is normally absent. It is known that REST levels can increase in later development or in response to cellular stress and target a subset of genes that are poised to respond to changes in REST levels [36,37].

## Nucleosome positioning at CTCF sites

Our analysis of chromatin states indicated a strong enrichment of nucleosomes at CTCF sites (chromatin state 8). To probe this interaction further, we derived 9,516 CTCF sites based on the presence of the CTCF-binding motif [38] in CTCF ChIP-seq data from H1 human embryonic stem cells (H1ESC). 67% of these CTCF sites are present within chromatin state 8, representing a 600-fold enrichment (Fig 2B). Again, there was a distinct peak in our small fragment data (122–137 bp) from pluripotent iPSC (Fig 5A), corresponding to CTCF binding. This peak was

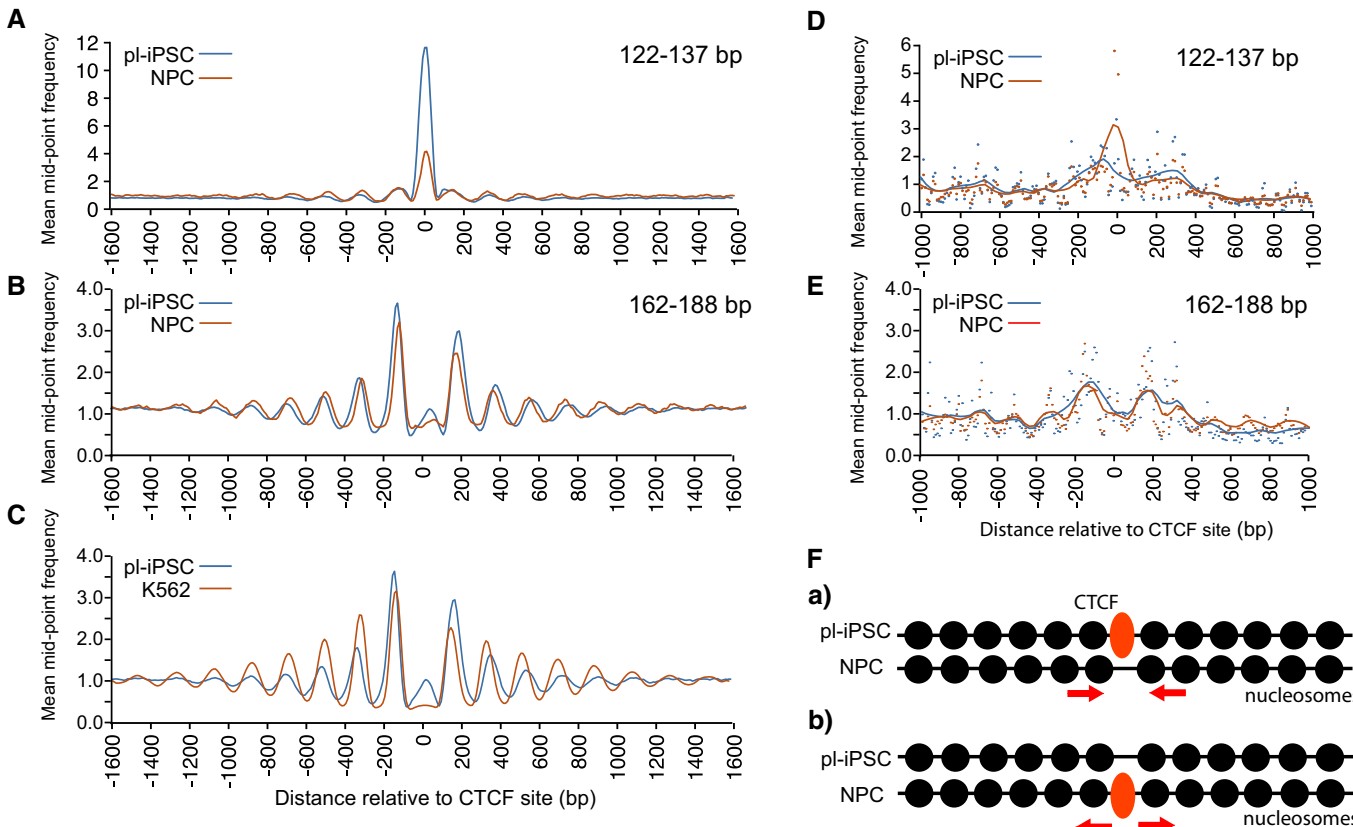

**Figure 5. Nucleosome patterning associated with the CTCF-binding sites.**

A  Average frequency distribution for sequence read mid-point data in the size range of 122–137 bp centred on the CTCF-binding site (n = 9,516). This shows a distinctive peak corresponding to the CTCF protein complex, which is present in pl-iPSC, but reduced in NPC.

B  Average frequency distribution for sequence read mid-point data centred on the CTCF-binding site in the size range of 162–188 bp, corresponding to larger nucleosome footprints, for pl-iPSC and NPCs. This shows that the pattern of nucleosome positioning is independent of the amount CTCF protein complex. Although positioned nucleosomes are retained flanking CTCF sites, their positions are shifted closer to the CTCF site and their spacing is altered.

C  Average frequency distribution for sequence read mid-point data centred on the CTCF site for pl-iPSC (162–188 bp) and K562 (total MNase-seq data) cells. Positioned nucleosomes are retained flanking CTCF sites in K562 cells, but with shifted positions and altered spacing.

D  Average frequency distribution for sequence read mid-point data in the size range of 122–137 bp for pl-iPSC and NPCs centred on the 2,522 CTCF-binding sites identified from NPC-specific ChIP-seq data. A distinctive peak corresponding to the CTCF protein complex is present in the NPC state, but reduced to background levels in pl-iPSC.

E  Average frequency distribution for sequence read mid-point data centred on the CTCF-binding sites in the size range of 162–188 bp, corresponding to larger nucleosome footprints, for pl-iPSC and NPC centred on the 2,522 CTCF-binding sites identified from NPC-specific ChIP-seq data. Nucleosome arrays are present in both cell states, independent of the CTCF protein complex. Nucleosomes in the CTCF sites in pl-iPSC tend to position closer to the CTCF site than in NPC. The data derived from the unique NPC CTCF sites have greater noise due to the smaller number of sites analysed.

F  Schematic of nucleosome patterning at the CTCF site in pl-iPSC and NPC. Nucleosome arrays are present in both cell states in the absences of CTCF complex. When CTCF is present, nucleosomes reposition to move a small distance from the CTCF-binding site and increase their spacing. Nucleosomes are represented by filled black circles.

significantly reduced in the NPCs, but not completely absent. Cluster analysis demonstrated that this peak did not arise from averaging a smaller number of sites with strong peaks, but in fact represented a decreased occupancy at all CTCF sites during the transition from pluripotent cells to NPC (Fig EV5). Nucleosome positioning data revealed an array of positioned nucleosomes flanking the CTCF site in pluripotent iPSC, and again, in a similar fashion to that seen for NRSF/REST, these were retained in NPCs, despite a substantial decrease in CTCF binding (Fig 5B). Further-more, our analysis of the MNase-seq dataset for K562 cells showed a similar nucleosome patterning surrounding the CTCF sites of these lymphoblastoid cells (Fig 5C) [19].

To investigate this further, we examined the pattern of nucleo-somes at CTCF sites occupied only in NPCs and not iPSC. By cross-comparison between our 122- to 137-bp small fragment data and CTCF ChIP-seq data from NPC (derived from H9 ESC), we identified 2,522 unique sites with CTCF bound only in the NPC state (Fig 5D). These sites again possessed positioned nucleosomes flanking the bound CTCF complex (Fig 5E). When the same sites were aligned using the nucleosome data from the pluripotent iPSC state, posi-tioned nucleosomes were also present in the absence of bound CTCF (Fig 5E). These results demonstrate that just as nucleosomes remain positioned after loss of CTCF following differentiation from pluripo-tent cells to NPC, the reciprocal relationship also occurs as positions where CTCF will bind after differentiation possess a pre-existing array of positioned nucleosomes.

Close inspection of the nucleosome positioning pattern showed that the spacing of nucleosomes both upstream and downstream of the CTCF site in NPC was altered and that they were positioned nearer to the CTCF site than in pluripotent cells (Fig 5B). A similar small positional shift is present in the K562 cells (Fig 5C). This indi-cates that minor re-positioning of nucleosomes flanking CTCF sites occurs during the early stages of neural cell differentiation. Data generated using the unique NPC CTCF sites also showed small changes in nucleosome position; however, in this case nucleosomes in the pluripotent state are generally closer to the CTCF site (Fig 5E). This suggests a mechanism where CTCF sites either retain or pre-form nucleosome arrays with low CTCF occupancy but are then slightly repositioned in the presence of high concentrations of the CTCF protein complex (Fig 5F).

A recent report using an alternative method based on Histone 3 ChIP-seq also showed a nucleosome array centred on CTCF of human myeloid leukaemia cells (HL60) and altered nucleosome spacing following retinoic acid-induced differentiation [39]. In *Dictyostelium*, loss of ChdC increases nucleosome spacing, indicat-ing a capacity to specifically regulate spacing within nucleosomes [12]. In mammals, CTCF has been shown to complex with CHD8, an orthologue of ChdC [40], suggesting a mechanism for nucleo-some translocation at these sites. Although apparently minor, small changes in nucleosome spacing are likely to have profound changes in the local chromatin structure [41] and given the role of CTCF in determination of the topological associated domain (TAD) struc-tures, this may lead to quite significant changes of high-level chro-matin architecture and hence gene regulation.

In conclusion, we have used genome-wide mapping of human iPSC to follow the dynamics of nucleosome positioning from the pluripotent state to an early stage of neuronal development. This provides the opportunity to follow changes across two developmental states and is particularly valuable to assess changes in gene activity. We observed a substantial increase in the number of positioned nucleosomes and re-positioning of nucleosomes during differentiation to NPC. Although the existence of small subgroups of genes where positioning is important cannot be excluded, there was no strong correlation between nucleosome posi-tioning and gene activity at the genome-wide level. In general, there are very few organised arrays of nucleosomes within the genomes of these human cell types, but remarkably, where nucleosome arrays do occur, they mark chromatin structures that are retained during cell differentiation. These may represent defined regulatory sites that control long-range chromatin changes and higher order 3D organisation [42]. We propose that these sites where arrays of posi-tioned nucleosomes are retained during development mark impor-tant regulatory nodes within the dynamic chromatin architecture—the 4D Nucleome.

## Materials and Methods

### Cell culture

The 34D6, male, human iPSC line (a gift from Prof Chandran, Edin-burgh) [43] was cultured in mTeSR1 (Stem Cell Technologies) following the manufacturer's instructions. Tissue culture plates (Nunclon, Invitrogen) were coated for at least 2 hours with Matri-gelTM (BD Biosciences, VWR) diluted 1:75 with DMEM/F12 (Invit-rogen). 34D6 iPSCs were plated at a density of $\sim 10^6$ cells/10-cm plate in complete mTeSR1TM medium containing 10 μM Y27632 (Tocris) and incubated at 37°C in a standard 5% $CO_2$, humidified incubator (Binder). To passage iPSC, cells were first treated with 10 μM Y27632 for 2 h, then washed with $Ca^{2+}/Mg^{2+}$-free PBS and cell colonies lifted from the plate by incubation with Dispase (Stem Cell Technologies) containing 10 μM Y27632 (Tocris). Colonies were fragmented by gentle trituration, collected by centrifugation (180 *g*) and re-suspended in medium for re-plating.

### Cell differentiation

For neural differentiation, a dual SMAD inhibition protocol was used [44]. 34D6 iPSCs were harvested as described above for iPSC passaging. 34D6 iPSC colony fragments were plated in non-adherent bacteriological grade culture dishes in ADF differentiation medium to allow for embryoid body formation [45]. ADF differentiation medium comprised advanced DMEM/F12TM medium (Invitrogen) supplemented with penicillin/streptomycin (5 μg/l, Invitrogen), L-glutamine (200 mM, Invitrogen), 1× lipid concentrate (Invitrogen), 7.5 μg/ml holo-transferrin (Sigma), 14 μg/ml insulin (Merck), and 10 μM β-mercaptoethanol (Sigma). Medium was supplemented with 10 μM Y27632 (Tocris) for the first 2 days, with 10 μM SB-431542 (Tocris) until day 4 of differentiation and 0.5 μM LDN193189 (Mil-tenyi) until day 8 of differentiation. Medium was changed every 2 days. At day 8, neuralised embryoid bodies were washed with $Ca^{2+}/Mg^{2+}$-free PBS and then dissociated by incubation at 37°C with Accutase (PAA laboratories). A single-cell suspension was obtained by gentle trituration and cells washed with ADF medium and harvested by centrifugation at 1,000 rpm. Neural progenitors were then plated onto tissue culture plates coated with 0.1 μg/ml

poly-L-lysine (Sigma) and 10 μg/ml laminin (Sigma) in ADF medium + 5 ng/ml FGF2. 34D6-derived neural progenitors were grown to sub-confluency and passaged once by dissociation with Accutase and re-plating. The 34D6 iPSC-derived neural progenitors were harvested for nucleosome preparation on day 16 of differentiation. The NPC population was validated using immunocytochemistry by checking for the presence of the NPC-specific markers, such as nestin and the loss of pluripotency markers, for example Oct-4. Greater than 90% of the differentiated population contained neural progenitors.

### In vivo MNase digestion of chromatin

Chromatin was prepared from three bioreplicates of human iPSC and the same cells differentiated to NPCs as described previously for the *S. cerevisiae* genome [11]. The cell membranes and nuclei were made permeable to MNase using NP-40 [46,47], and each bioreplicate was treated by *in vivo* digestion with 300 U/ml MNase at room temperature for 4 min. DNA fragments were purified from each MNase treated sample, and then, the samples for each cell type were pooled in equimolar amounts. DNA extracted from chromatin samples was size-fractionated on agarose gels. 25–35 μg of DNA less than 300 bp was size-selected for each cell type.

### Paired-end mode DNA sequencing

All of the DNA fragments less than 300 bp in size were utilised for paired-end mode sequencing by Source BioScience (http://www.sourcebioscience.com/) on an Illumina HiSeq 2000 platform (HiSeq) using a read-length of 50 bp. Eight flow cells were used for each cell type to obtain a sufficient depth of coverage of the human genome. A standard Illumina paired-end mode sequencing protocol was used, apart from the omission of the nebulisation step and the addition of a further gel purification step to eliminate any excess concatenated linkers after the ligation of linker DNA to the sample. Base calling and quality control of the sequencing data were performed using Real Time Analysis (RTA) 1.09, CASAVA 1.8 software.

### Alignment of paired-end reads to the genome

A total of 3.4 and 3.0 billion paired-end reads were obtained in fastq format from iPSC and NPC, respectively, and aligned to the human genome assembly hg19 using Bowtie version 0.12.8 [48]. The command line options for bowtie were as follows: bowtie -v 3 –trim3 14 –maxins 5000 –fr -k 1 –best -p 12.

### Creation of nucleosome maps

Subsequent data processing to create nucleosome maps from iPSC and NPC was undertaken in a similar manner to the method previously described for the *S. cerevisiae* genome [11].

### Data validation and normalisation

The number of reads obtained for each human autosome in each cell type was determined and utilised for further analysis. To compensate for the slight genome-wide difference in total read counts between the two cell types, the read counts for the NPC autosomal genome were multiplied by the ratio of the total aligned reads in the autosomal genomes of iPSC v NPCs, which was 1.117.

### Read mid-point frequency distributions

Paired-end reads obtained from Bowtie alignments in SAM format were sorted into separate chromosome-specific files. To represent a unique position for each paired-read, the genomic position of the mid-point of the insert DNA was calculated. The reads were separated and filtered into three size classes: 112–137 bp, 138–161 bp (nucleosomes), 161–188 bp. The frequency distribution for the mid-point position of the sequencing reads was derived at 10-bp resolution, and the data were smoothed using a 3-bin moving average. Frequency distributions were output as chromosome-specific files in .sgr format: chromosome identification: chromosomal location of the start of each 10-bp bin: frequency of the paired-read mid-point values that fall within that 10-bp bin.

### Published human nucleosome maps

Published human nucleosome maps [19] from the K562 chronic myelogenous leukaemia (Cml) cell line [49] and from the B-lymphoblastoid cell line GM12878 (Coriell Biorepository) were converted from bigwig to bedgraph format, and the chromosome-specific bedgraph files were then converted to .sgr files by binning the data into 10-bp bins and calculating a 3-bin moving average exactly as described for the iPSC and NPC data above. The final re-processed maps were validated as for the iPSC and NPC maps, generating the total numbers of aligned paired-end reads for each genome.

### Locating patterns of positioned nucleosomes

In order to locate and quantify the number of highly positioned nucleosomes, a heuristic peak-finding algorithm was developed. Peaks were defined as three consecutive 10-bp bins where the value of the paired-read mid-point frequency in the central 10-bp bin was between 30 and < 1,000. The lower threshold for the paired-read mid-point frequency values for the bins either side of the central bin was set at two (the "noise" threshold). The upper threshold for each of the paired-read mid-point frequency values for the bins either side of the central bin was less than the paired-read mid-point frequency in the central bin. The upper threshold was chosen to exclude regions of the genome with high frequencies of paired-read mid-point values found at the ends of chromosomes and at runs of repeats and at centromeres. Peaks in the genomic distribution of sequence read mid-points were given explicit genome positions using PeakFinder for all of the nucleosome maps: iPSC, NPC and K562.

### Detecting nucleosome arrays

An in-house python script was used to calculate the distance between the genomic locations of all highly positioned nucleosomes across the genome for each cell type, iPSC, NPC and K562. Positioned nucleosomes 150–200 bp apart were located, and those existing as singletons or in arrays of 2, 3 or more nucleosomes

were counted. In-house Perl scripts were used to compare the genome-wide overlap in the locations of positioned nucleosomes within ± 10 bp in iPSC and NPC (Fig 1C).

### Chromatin states

Chromatin state data file: wgEncodeBroadHmmH1hescHMM.bed containing chromatin state information derived from H1ESCs was downloaded from http://genome.ucsc.edu/cgi-bin/hgFileUi?db= hg19&g=wgEncodeBroadHmm. For iPSC, we aligned the genomic positions of the peaks in the nucleosome map determined using the heuristic PeakFinder with the chromatin state regions determined in H1ESC. Thus, we calculated the genome-wide total for the number of nucleosomes in the following chromatin states [20,21]: active promoter = state 1; insulator = state 8; repressed = state 12; heterochromatin = state 13. We calculated the expected number of nucleosomes in any particular chromatin state, assuming that positioned nucleosomes are dispersed randomly across the genome (Table EV3). Similarly, we determined the chromatin state for the CTCF sites we derived (using the H1ESC ChIP-seq data combined with the CTCF-binding motif) and calculated the expected chromatin states for CTCF sites assuming that their distribution across the genome is random (Table EV4).

### Open chromatin

Published open chromatin data [23] derived from ATAC-seq studies of two undifferentiated human iPSC were downloaded from GEO (https://www.ncbi.nlm.nih.gov/geo/query/acc.cgi?acc=GSE85330).

We aligned the genomic positions of the peaks in the nucleosome map determined using the heuristic PeakFinder with the autosomal ATAC-seq regions from each of two replicate cell lines C15_0_1_ATAC-seq (GSM2264802) and C15_0_2_ATAC-seq (GSM2264803) using in-house python scripts. The results of these analyses from the two replicates were similar. Figure 2C and D show the results of the analysis from ATAC-seq regions from C15_0_1_ATAC-seq (GSM2264802). We calculated the expected number of nucleosomes and CTCF sites in open chromatin assuming that nucleosomes are randomly distributed across the genome.

### Construction of genomic feature lists

The positions of transcriptional start sites for all of the full-length transcripts for human genome assembly GRCH37/hg19 were derived as follows. Track = Gencode Genes v17, table = basic, was downloaded from the UCSC genome browser (contains 94,151 TSS). From this, a list of non-redundant, strand-specific autosomal TSS was derived ($n = 83,179$) and a list of non-redundant, strand-specific autosomal TSS from protein-coding genes ($n = 66,047$) using in-house scripts was derived. Expression data from ESCs and ESCs differentiated to the N2 stage of development [26] were used to generate two lists of genes expressed specifically in a) ESCs ($n = 3,833$) and b) N2 cells ($n = 2,082$). The minimum expression threshold was 0.1 (normalised RPKM), and the difference in expression was at least twofold difference between ESC and NPC in each case. TSS of genes expressed specifically in (i) ESC and (ii) N2 cells were tested for the presence of positioned nucleosomes within ± 300 bp of the TSS by

mapping the genomic locations of highly positioned nucleosomes generated by the PeakFinder tool to a window ± 300 bp of the TSS in each case.

### Transcription factor binding motifs

The genomic positions of the consensus binding sequence for each transcription factor were extracted from FASTA files from human genome assembly GRCH37/hg19 using in-house Perl scripts. FASTA files were downloaded from: http://hgdownload.cse.ucsc.edu/golde nPath/hg19/chromosomes/.

The consensus binding motif used in this study for each transcription factor is shown in Table EV5. The consensus binding motif for YY1, M1, from factorbook derived from H1ESC ChIP-seq data was used (http://www.factorbook.org) [50]. The ATF2 CRE-binding motif was taken from Hai *et al* [51]. The PAX6 consensus binding sequence was derived using (ChIP) in ES-derived neuroectodermal cells (NECs) [52]. CTCF-binding positions in pluripotent cells were derived by mapping the consensus binding motif derived by Ong *et al* [38] with H1ESC ChIP region data from the Broad Institute downloaded from the UCSC genome browser, file wgEncodeAwgTfbsBroadH1h-escCtcfUniPk.narrowPeak (UCSC accession wgEncode EH000085). NPC-specific CTCF-binding sites were derived by matching using the locations of the peaks in the protected DNA fragments in the size range 122-137bp in NPCs with H9ESC-derived NPC ChIP data from the ENCODE project (https://www.encodeproject.org/files/ ENCFF796YPF), from data with conservative idr thresholded peaks using in-house python scripts. RE1 sites were derived by mapping the consensus binding motif derived by Bruce *et al* [32] with ChIP data that were derived from the ENCODE database [34]. ChIP data were downloaded using the UCSC genome browser from human genome assembly GRCH37/hg19 Group = regulation, Track name = TXnFactorChIP, Table = wg EncodeRegTfbsCLusteredv2 from uniform processing of data from the Jan. 2011 ENCODE data freeze.

### SiteWriter

The in-house Perl script SiteWriter [11] was used to construct average frequency distributions of the sequencing read mid-point values at and surrounding genomic feature loci within a user-defined window, for example at transcription factor binding sites, and surrounding positioned nucleosomes (Fig 1D). The output from the SiteWriter script comprises two files: (i) a CFD.txt file of the normalised average frequency values. The values are normalised by dividing the frequency value in each bin by the number of bins specified in the user-defined window. (ii) a C3.txt file which contains a matrix of locally normalised dyad frequency values for every bin position. The C3.txt file data were used in cluster analysis.

### Cluster analysis

Cluster analysis was undertaken in R, using the Canberra method to generate a distance matrix from the C3.txt file from the output of the SiteWriter script. Dendrograms generated by hierarchical agglomerative clustering were used to determine the number of groups to use in k-means clustering. Cluster data for genomic features were used to construct average frequency distributions of

the read mid-point values at and in the bins surrounding genomic feature loci within a user-defined window for selected clusters.

## Data availability

Raw and processed sequencing read data are available on GEO: https://www.ncbi.nlm.nih.gov/geo/query/acc.cgi?acc = GSE117870

All the scripts used in this analysis are available on request.

**Expanded View** for this article is available online.

## Acknowledgements

The research reported in the manuscript was supported by grant from the Waterloo Foundation to A.J.H and a BBSRC-funded Daphne Jackson fellowship awarded to J.C.H.

## Author contributions

JCH, NDA and AJH designed the experiments; JCH and NAK generated the data; JCH and AJH performed the analysis and wrote the manuscript.

## Conflict of interest

The authors declare that they have no conflict of interest.

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
