## [Review Process File · EMBO Reports]

Nucleosome dynamics of human iPSC during neural differentiation

Janet C Harwood, Nicholas A Kent, Nicholas D Allen and Adrian J Harwood.

Review timeline:

Submission date:	24 th August 2018
Editorial Decision:	24 th September 2018
Revision received:	18 th January 2019
Editorial Decision:	1 st February 2019
Revision received:	29 th March 2019
Accepted:	2 nd April 2019

Editor: Esther Schnapp

Transaction Report:

1st Editorial Decision

24th September 2018

As you will see, both referees acknowledge that the findings are potentially interesting. However, they both also point out technical concerns that preclude a solid interpretation of the experimental evidence provided. I think that all referee concerns should be addressed, and if you think that you can do so, I would like to invite you to revise your manuscript along the lines suggested by the referees. In this case, please also address all referee concerns in a complete point-by-point response.

Acceptance of the manuscript will depend on a positive outcome of a second round of review. It is EMBO reports policy to allow a single round of revision only and acceptance or rejection of the manuscript will therefore depend on the completeness of your responses included in the next, final version of the manuscript.

REFeree REPORTS.

Referee #1:

In this manuscript entitled: "Nucleosome dynamics of human iPSC during the early stages of Neurodevelopment", Harwood and colleagues have carried out micrococcal digestion followed by sequencing (MNase-seq) on human iPSCs and the in vitro differentiated NPCs to map nucleosome positioning during neurodevelopment. The authors have also compared their MNase-seq data with published data generated from K562 cell line. The main conclusion of this study is that the undifferentiated iPSCs contain 8-fold less positioned nucleosomes compared to that of NPCs and K562. Interestingly, despite being sparse, the majority of these well positioned nucleosomes are within insulators and not close to TSS. Unexpectedly, the authors report that there is a fraction of nucleosomes that do not change positioning despite loss of CTCF and NRSF/REST binding upon differentiation.

Overall, I find the manuscript to be well written and the data well-presented. The main issue with this study is the extent of MNase digestion used to map nucleosomes in the two cell types which is relatively high. This amount of MNase have been classically used to maximize mono-nucleosomes yield from bulk chromatin. But it is becoming appreciated in the field that the amount of MNase have profound effects on the type of nucleosomes being mapped (Iwafuchi-Doi et al., 2016; Mieczkowski et al., 2016; Mueller et al., 2017). Thus, under high-level MNase digestion reaction (similar to the one used in this study), nucleosomes in open chromatin are preferentially destroyed over nucleosomes in closed chromatin. Because the chromatin in undifferentiated iPSCs is significantly more open than that of their differentiated NPCs counterparts, nucleosomes will be over-digested in iPSCs. This may well reflect the low number of the poisoned nucleosomes (8 fold less) identified in iPSCs. Therefore, the authors are not solely measuring the dynamics of nucleosome positioning during differentiation but also the sensitivity to MNase digestion. The authors tried to argue that the chromatin structure is not changed by looking at the distribution of nucleosome array patterns (Figure 1D). However, this is only considering the sequenced data which was generated from size selected DNA fragments under a single MNase concentration. The authors need to show the total MNase digestion pattern of the iPSCs genome as compared to NPCs using carefully titrated MNase amounts. This will show the differential effects of MNase digestion on the two cell types. To circumvent this confounding problem, it is currently common practice to sequence using various amounts of MNase. Nevertheless, this study is still valid if the authors clearly state the limitations of their high MNase digestion approach and address the additional concerns below:

Other concerns:

- The nucleosome density may be different between iPSCs and NPCs. Is the amount of chromatin associated histones the same between the two cell types?
- Venn diagram in Figure 1B: the cell types are not indicated.
- Figure 2C: The highlighted positioned nucleosome in the NANOG locus seems to be (-1)relative to TSS not (+1)as indicated.
- Figure 2E: Black and grey bars indicate iPSCs and NPCs. How come the underlined black and grey bars representing active and repressed genes are also underlined as a single cell type?
- It seems that the authors used RE1 and CTCF sites mapped in humans ESCs, what happens to RE1 and CTCF sites that are bound in NPCs? Do these sites gain nucleosome positioning or nucleosomes were pre-positioned in iPSCs prior to occupancy by CTCF and RE1.

References:

- Iwafuchi-Doi, M., Donahue, G., Kakumanu, A., Watts, J.A., Mahony, S., Pugh, B.F., Lee, D., Kaestner, K.H., and Zaret, K.S. (2016). The Pioneer Transcription Factor FoxA Maintains an Accessible Nucleosome Configuration at Enhancers for Tissue-Specific Gene Activation. *Mol Cell* 62, 79-91.
- Mieczkowski, J., Cook, A., Bowman, S.K., Mueller, B., Alver, B.H., Kundu, S., Deaton, A.M., Urban, J.A., Larschan, E., Park, P.J., et al. (2016). MNase titration reveals differences between nucleosome occupancy and chromatin accessibility. *Nat Commun* 7, 11485.
- Mueller, B., Mieczkowski, J., Kundu, S., Wang, P., Sadreyev, R., Tolstorukov, M.Y., and Kingston, R.E. (2017). Widespread changes in nucleosome accessibility without changes in nucleosome occupancy during a rapid transcriptional induction. *Genes & Development* 31, 451-462.

Referee #2:

The current "textbook" view of nucleosome positioning within eukaryotic cells is that nearly every gene transcribed by RNAPII contains a nucleosome depleted region upstream of the transcription start site (TSS), followed by a highly positioned nucleosome near the TSS (the +1 nucleosome). Furthermore, there is a general feel that this promoter proximal, positioned nucleosome should be flanked by short, positioned nucleosomal arrays. This view of nucleosome positioning as it relates to gene expression is based initially on high resolution nucleosome mapping studies in budding yeast, but similar results have been found in nucleosome mapping studies in *C. elegans* (*Genome Res.* 2008 18:1051-63), *Drosophila* (*Nature.* 2008 453:358-62), and mouse ESCs (*Nat. Struct. Mol. Biol.*, 19 (2012), pp. 1185-1192; *Dev. Cell* 2014 30:11-22). In addition to mapping nucleosomes surrounding promoter regions, several studies have also demonstrated positioning of nucleosome

that surround binding sites for CTCF (Nat. Struct. Mol. Biol., 19 (2012), pp. 1185-1192; Dev. Cell 2014 30:11-22), indicating that CTCF organizes an array of positioned nucleosomes surrounding its binding sites.

In this current manuscript, Harwood and colleagues generate high resolution nucleosome maps by paired end sequencing of MNase digested hiPSCs and differentiated NPCs. They report that only 3% of all nucleosomes are highly positioned in NPCs and 8x fewer nucleosomes are highly positioned in undifferentiated hiPSCs. Surprisingly, the authors do not appear to detect a general enrichment of positioned nucleosomes near the TSS of active or inactive genes, though this reviewer was surprised to see no analysis of nucleosome positioning globally aligned to TSSs. In contrast, the authors focus on nucleosome positioning flanking binding sites for CTCF and for NRSF/REST. Here they find positioned nucleosome arrays flanking these factors, and the positioning persists in the differentiated NPCs where these factors are either missing (NRSF/REST) or decreased (CTCF).

It appears that a primary conclusion of this work is that most nucleosomes in either hiPSCs or NPCs are not highly positioned. However, it is not clear if this is a new observation. Previous nucleosome mapping studies have focussed primarily on nucleosome positioning as it pertains to promoters and enhancers, and here there seems to be a general consensus in the literature that positioned nucleosomes are a hallmark of these elements. This manuscript would be improved by more detailed comparison of their dataset with the numerous other studies that have investigated nucleosome positioning in higher eukaryotes, especially those that have specifically assayed ESCs and differentiated ESCs (e.g. the study from the Rando group; Dev. Cell 2014). Are NFR regions detected? Are nucleosomes generally positioned near TSSs? If not, why is this data different from previous work? Is the only distinction that this current work has been done in hiPSCs rather than mouse ESCs? Or is there another distinction?

What is also not clear is what is meant by the term "highly positioned". This needs a quantitative definition. The worry is that the stringency of this term is too high, eliminating nucleosomes that are "highly" positioned but might vary in a 10 bp window or less.

The authors need to show representative MNase digestions that are used for the library constructions. There is a wealth of literature detailing how over or under digestions with Mnase can skew results. Indeed, this is why the Rando group sequenced the entire MNase digestion rather than gel purifying a particular size range.

1st Revision - authors' response

18th January 2019

EMBOR-2018-46960V2

Reply to Reviewer's comments

General

Both reviewers were positive about our manuscript and made some extremely helpful comments and proposed revisions, particularly in how our results may integrate with current thinking in this research area. In part, the reviewer's comments arise from the brevity of our original submission. We used the "initial submission" process with a manuscript written for a journal with a much shorter format and style, and this limited the information that we could include. In revising this manuscript, we have been able to use the longer EMBO Report format to address the majority of the reviewer's concerns, and still remain within the character limit for a EMBO Reports scientific report. In addition, we have conducted new analysis of our data to address the remaining comments and also added a few additional analyses relevant to the general discussion.

We therefore have undertaken major revisions to address **all** of reviewer's points, and hope that the revised manuscript is now suitable for publication.

Specific comments.

Referee #1:

1. **Comment:** The main conclusion of this study is that the undifferentiated iPSCs contain 8-fold less positioned nucleosomes compared to that of NPCs and K562. Interestingly, despite being sparse, the majority of these well positioned nucleosomes are within insulators and not close to TSS. Unexpectedly, the authors report that there is a fraction of nucleosomes that do not change positioning despite loss of CTCF and NRSF/REST binding upon differentiation. Overall, I find the manuscript to be well written and the data well-presented.

Reply: Although the reviewer summarises what we observe, they may have slightly misinterpreted our conclusions. We have addressed this issue with modifications throughout the text and additional analysis to clarify our conclusions. To summarise our conclusions, we have used differentiation of the same cells from the pluripotent to NPC state to follow the changes of nucleosome pattern that occur during early stages of neurodifferentiation and how they affect gene regulation. In fact, we do see positioned nucleosomes at the TSS, but their presence does not correlate with whether the gene is active or not. In contrast we find extended nucleosome arrays at RE1 (the NRSF/REST binding site) and CTCF sites, which act on gene regulation at long distance. Unexpectedly, these sites have positioned nucleosomes independent of the bound regulatory complex. We propose a scenario where rather than acting locally to control gene expression, nucleosome positioning can generate sites within chromatin that lead to long range changes in gene expression. We think that this is important to report, as often nucleosome positioning associated with gene expression is conveyed simply in terms of accessibility to DNA, our results suggest additional modes of action via creation of chromatin sites that may mediate long range regulatory interactions.

2. **Comment:** The main issue with this study is the extent of MNase digestion used to map nucleosomes in the two cell types which is relatively high. This amount of MNase have been classically used to maximize mono-nucleosomes yield from bulk chromatin. But it is becoming appreciated in the field that the amount of MNase have profound effects on the type of nucleosomes being mapped (Iwafuchi-Doi et al., 2016; Mieczkowski et al., 2016; Mueller et al., 2017). Thus, under high-level MNase digestion reaction (similar to the one used in this study), nucleosomes in open chromatin are preferentially destroyed over nucleosomes in closed chromatin. Because the chromatin in undifferentiated iPSCs is significantly more open than that of their differentiated NPCs counterparts, nucleosomes will be over-digested in iPSCs. This may well reflect the low number of the positioned nucleosomes (8 fold less) identified in iPSCs. Therefore, the authors are not solely measuring the dynamics of nucleosome positioning during differentiation but also the sensitivity to MNase digestion.

Reply: We fully understand this point and we have made a number of revisions to the text to address it.

i) We have included a detailed discussion in the Introduction (3rd paragraph), incorporating the suggested references. This gives the background to our approach and how it compares with the MNase titration methods.

ii) In the Results (paragraph 3), we provide additional evidence that the 8-fold decrease in positioned nucleosomes in the pluripotent state is not due to higher rates of MNase digestion.

iii) We include an analysis of open chromatin and show that our positioned nucleosomes are in fact enriched in open chromatin, and not depleted.

3. **Comment:** The authors tried to argue that the chromatin structure is not changed by looking at the distribution of nucleosome array patterns (Figure 1D). However, this is only considering the sequenced data which was generated from size selected DNA fragments under a single MNase concentration. The authors need to show the total MNase digestion pattern of the iPSCs genome as compared to NPCs using carefully titrated MNase amounts. This will show the differential effects of MNase digestion on the two cell types. To circumvent this confounding problem, it is currently common practice to sequence using various amounts of MNase. Nevertheless, this study is still valid if the authors clearly state the limitations of their high MNase digestion approach and address the additional concerns below:

Reply: We have addressed this issue in a number of ways

i) We have better defined our experiments to explain why we are investigating mono-nucleosome positioning.

ii) Included an EV figure and a new Fig 1B to show the effects of MNase digestion on both pluripotent and NPC chromatin

4. Comment: The nucleosome density may be different between iPSCs and NPCs. Is the amount of chromatin associated histones the same between the two cell types?

Reply: The new Fig EV1 shows that based on the nucleosomal DNA (approx. 150 bp band on an agarose gel), nucleosome densities are similar between both cell states.

5. Comment: Venn diagram in Figure 1B: the cell types are not indicated.

Reply: This error has now been corrected

6. Comment: Figure 2C: The highlighted positioned nucleosome in the NANOG locus seems to be (-1) relative to TSS not (+1) as indicated.

Reply: On a gene-by-gene basis, we find many genes, which appear to possess a positioned nucleosome only at the -1 position. We have now included our global analysis of the TSS in human cells. When the data is combined across the 83,000 listed TSS positions, a pattern of -1, +1 and +2 can be discerned, but clearly there is a lot of variation at the level of individual genes.

7. Comment: Figure 2E: Black and grey bars indicate iPSCs and NPCs. How come the underlined black and grey bars representing active and repressed genes are also underlined as a single cell type?

Reply: This figure was complicated and confusing to the reader. It has now been replaced by a simpler (and clearer) version.

8. Comment: It seems that the authors used RE1 and CTCF sites mapped in humans ESCs, what happens to RE1 and CTCF sites that are bound in NPCs? Do these sites gain nucleosome positioning or nucleosomes were pre-positioned in iPSCs prior to occupancy by CTCF and RE1.

Reply: It is not possible to do this for RE1 as NRSF/REST is not bound to the site in NPC, however this is a very interesting question with regard to CTCF. We have now included a new analysis of our data to address this point. This was a very good point suggested by the reviewer and greatly strengthens our observations for the nucleosome pattern at CTCF, and potential mechanisms.

It also proved to be a challenge to identify CTCF sites bound only in NPC cells as they did not appear to possess the CTCF consensus site that works in the pluripotent state. However, by cross comparison of our small fragment data with the Encode NPC ChIP-seq data, we were able to identify 2,522 CTCF sites uniquely bound to the CTCF complex in NPC. The smaller data set possesses slightly higher background noise than the pluripotent set using 9,478 sites, however clearly shows a nucleosome array at the bound sites in NPC, but importantly that the array pre-exists in the pluripotent state prior to CTCF binding. It also shows a trend for the change in nucleosome positioning as CTCF binds.

Referee #2:

1. Comment: Surprisingly, the authors do not appear to detect a general enrichment of positioned nucleosomes near the TSS of active or inactive genes, though this reviewer was surprised to see no analysis of nucleosome positioning globally aligned to TSSs.

Reply: We have now included our data for the global alignment at the TSS (now Fig 3C). As described above, at this whole genome level positioned nucleosomes and a NFR can be seen, although there is considerable variation on a gene by gene basis. The important point is that the poor correlation between the presence of positioned nucleosomes and gene activity. We have now stressed this point in the text.

2. Comment: In contrast, the authors focus on nucleosome positioning flanking binding sites for CTCF and for NRSF/REST. Here they find positioned nucleosome arrays flanking these factors, and the positioning persists in the differentiated NPCs where these factors are either missing (NRSF/REST) or decreased (CTCF).

It appears that a primary conclusion of this work is that most nucleosomes in either hiPSCs or NPCs are not highly positioned. However, it is not clear if this is a new observation. Previous nucleosome mapping studies have focussed primarily on nucleosome positioning as it pertains to promoters and enhancers, and here there seems to be a general consensus in the literature that positioned nucleosomes are a hallmark of these elements.

Reply: Although the very low proportion of positioned nucleosomes in mammalian and human cells is a surprise in comparison to organisms with small genome sizes, it has indeed been found before. We point this out in the manuscript, and even use some of this earlier data in our analysis. What we have done in this paper is to use stem cell differentiation to follow the same cells as they differentiate towards the neuronal cell fate. This allows us to track changes in nucleosome position in the same cell population. Our results suggest that there is a poor correlation between changes in nucleosome position and gene activity. Instead, we make a number of observations about the sites where nucleosome arrays are present in both cell states. These do not appear to be at promoters or most transcription factor binding sites but are present at RE1 and CTCF sites that mediate long range changes in gene regulation. Based on these observations we propose that nucleosome positioning can serve a role in creation of chromatin sites where long-range gene regulatory proteins bind. This effectively gives nucleosome positioning an upstream regulatory role in gene expression that differs from a conventional view where nucleosome placement acts as a downstream effector altering gene expression via DNA accessibility.

3. Comment: This manuscript would be improved by more detailed comparison of their dataset with the numerous other studies that have investigated nucleosome positioning in higher eukaryotes, especially those that have specifically assayed ESCs and differentiated ESCs (e.g. the study from the Rando group; Dev. Cell 2014). Are NFR regions detected? Are nucleosomes generally positioned near TSSs? If not, why is this data different from previous work? Is the only distinction that this current work has been done in hiPSCs rather than mouse ESCs? Or is there another distinction?

Reply: In general we have included as much cross comparison with previous relevant data as possible. We show that the data we have generated in our neurodevelopmental cell mode is compatible with that previously published for other cell types. This is important given our novel observations concerning gene regulation and the presence of nucleosome arrays in the absence of regulatory complexes. As suggested we have now included more of our analysis, for example, our demonstration of genome-wide consensus nucleosome pattern that shows remarkable similarities to that seen in yeast and Dictyostelium, as well as that shown for human cells by the Rando group.

Comment: What is also not clear is what is meant by the term "highly positioned". This needs a quantitative definition. The worry is that the stringency of this term is too high, eliminating nucleosomes that are "highly" positioned but might vary in a 10 bp window or less.

Reply: We have included more details in the text about how we define "highly positioned".

Comment: The authors need to show representative MNase digestions that are used for the library constructions. There is a wealth of literature detailing how over or under digestions with Mnase can skew results. Indeed, this is why the Rando group sequenced the entire MNase digestion rather than gel purifying a particular size range.

Reply: We have included this data as Fig EV1

Thank you for the submission of your revised manuscript. While both referees support its publication now, they do raise a few more points that need to be successfully addressed before we can proceed with the official acceptance of your study.

REFEREE REPORTS

Referee #1:

In this revision, Hartwood and colleagues have addressed many of the original concerns as listed in the rebuttal letter. However, there are still few points that I highly recommend being addressed prior to the publication of this manuscript in EMBO reports.

1- Paragraph 3: The statement: "Studies in a range of mouse and human cells..." missing references.

2- The major conclusion of this study as shown in Figure 1B, is that iPSCs show less positioned nucleosomes than NPCs. The authors address the previous concerns that this is not due to MNase over-digestion by measuring sub-nucleosome fraction enrichment (Figure V1C). However, the authors need to rule out that this can also be due to an overall less-nucleosome level in iPSC (less histone proteins) before they can conclude that this is solely due to less positioned nucleosomes.

3- In Fig 1D and 1E, the authors show that most positioned nucleosomes exist as singletons and not arrays. This is not necessarily true, as only positioned nucleosomes are considered in these analyses. Positioned nucleosomes can be in an array within non-positioned nucleosomes. First in Fig. 1D, by measuring MNase-enrichment centred around positioned nucleosomes the flanking nucleosomes will not show an enrichment if the distance to the centered nucleosome is highly variable as the signal will be diffused across the region examined. Second in Fig. 1E, by counting the number of nucleosomes within 150-200bp from the centre (dyad), i.e. the actual distance between the nucleosomes (between exit and entry) that the authors investigated would be from 0-50 bp, which is less than average (50-70 bp). It will be more informative and less biased, if the authors plot the distribution of distance of each positioned nucleosome to the nearest positioned nucleosome. This would show how close positioned nucleosomes are to each other and not falsely display these positioned nucleosomes as singletons by ignoring the non-positioned nucleosomes.

4- Figure 2C and 2D don't exist but cited in the text.

5- The authors used ATAC-seq to show that positioned nucleosomes are enriched within open chromatin, addressing one of the previous concerns. However, the authors have to make sure that the ATAC-seq DNA fragments considered for this analysis are smaller than nucleosomes. As it is well known that ATAC-seq also map nucleosome positions and not solely represent open chromatin (Buenrostro et al Nat Methods 2013). This is particularly important if the tagmented DNA is not size-selected prior to sequencing as was done with the published data used for this study.

6- The authors have shown the distribution of positioned nucleosomes centred around TSS of selected loci in Figure 3A and 3B, and genome-wide figure 3C. The authors have also shown the difference between the total enrichment of positioned nucleosomes within 300bp of TSS with relation to gene activity (Figure 3D). I recommend that the authors also show the difference in the distribution of positioned nucleosomes in relation to TSS of active vs non-active genes (the same gene sets in figure Figure 3D).

7- The authors investigated the positioned nucleosome arrays that are enriched around RE1 (n=871) and CTCF (n=9,516) binding sites, which are in total around 10,000 sites in iPSCs. Taken that about 65% of these sites are surrounded positioned nucleosomes. One should expect for each of the 6,500 of RE1 and CTCF binding sites, there are more than three positioned nucleosomes. However, the authors made a major point that less than 0.1% of the positioned nucleosomes are within positioned arrays (more than 3 nucleosomes), which should be around 48 sites (0.1% of the

total 48,840 positioned nucleosomes in iPSCs fig 1E). This doesn't make sense. The authors need to clarify this discrepancy.

8- The nucleosomes around the CTCF sites occupied only in NPC cells and not iPSC shown in figure 5E are clearly different from those shown in Figure 5B. The peaks are sharper and less positioned. The authors have completely ignored this obvious difference and stated as if the nucleosome positioning show similar pattern in both cases to strengthen their conclusion on the uncoupling between CTCF presence and nucleosome positioning. This issue have to be addressed in the revision.

Referee #2:

The authors have done an excellent job at responding to my previous comments. In particular, the inclusion of data for nucleosomes surrounding gene TSS regions makes a solid point.

I do have one main point that the authors should discuss:

Only 32% of the positioned nucleosomes in pluripotent cells maintain positioning in NPCs, and also NPCs have 8x more positioned nucleosomes. It is not clear from the analysis/discussion where these new nucleosomes are located in the NPCs. Much of the discussion seems more focused on the small percentage of common, positioned nucleosomes.

Minor point:

In small genome organisms, positioned nucleosomes along gene bodies are likely due to boundary effects - NDRs flank the small coding regions. This provides a simple explanation for the lack of positioned nucleosomes along gene bodies in metazoans.

2nd Revision - authors' response

29th March 2019

We are very pleased to see that we have satisfied both reviewers with our revised manuscript. A number of additional points were raised by the reviewers, particularly Reviewer 1, and a number of editorial modifications suggested. I am pleased to say that we have been able to accommodate all of these additional suggestions (see below) and believe that our manuscript should be acceptable for publication.

REFEREE REPORTS.

Referee #1:

1. **Comment:** Paragraph 3: The statement: "Studies in a range of mouse and human cells..." missing references.

Reply: Relevant references now included.

2. **Comment:** The major conclusion of this study as shown in Figure 1B, is that iPSCs show less positioned nucleosomes than NPCs. The authors address the previous concerns that this is not due to MNase over-digestion by measuring sub-nucleosome fraction enrichment (Figure V1C). However, the authors need to rule out that this can also be due to an overall less-nucleosome level in iPSC (less histone proteins) before they can conclude that this is solely due to less positioned nucleosomes.

Reply: We do not think that that there is a major difference in the actual total number of nucleosomes, and certainly not an 8-fold difference (12.5% of the number in differentiated cells). It is important to note that the changes we are describing here are in the number of **positioned**

nucleosomes. These are nucleosomes that are found in the same sequence position in the majority of cells in the population, in contrast to the large majority of nucleosomes, which have different positions from one cell to the next. As all nucleosomes have the same footprint (approx. 150 bp of DNA) regardless of whether they are positioned or not, then the mononucleosome intensities on the gels shown in Fig EV1 are a good estimate of the total number of nucleosomes. Quantification of the band intensities of MNase digests from pluripotent and NPC chromatin, show approximately the same amounts. In fact, if anything the pluripotent cells actually have a slightly higher amounts of mononucleosome DNA, consistent with the larger size distribution of the pair-ended sequence reads, shown in Fig EV 1B&C. To make this point clearer, we have modified the relevant text (first paragraph of Results (p6) and Fig EV1 figure legend).

3. Comment: In Fig 1D and 1E, the authors show that most positioned nucleosomes exist as singletons and not arrays. This is not necessarily true, as only positioned nucleosomes are considered in these analyses. Positioned nucleosomes can be in an array within non-positioned nucleosomes. First in Fig. 1D, by measuring MNase-enrichment centred around positioned nucleosomes the flanking nucleosomes will not show an enrichment if the distance to the centred nucleosome is highly variable as the signal will be diffused across the region examined. Second in Fig. 1E, by counting the number of nucleosomes within 150-200bp from the centre (dyad), i.e. the actual distance between the nucleosomes (between exit and entry) that the authors investigated would be from 0-50 bp, which is less than average (50-70 bp). It will be more informative and less biased, if the authors plot the distribution of distance of each positioned nucleosome to the nearest positioned nucleosome. This would show how close positioned nucleosomes are to each other and not falsely display these positioned nucleosomes as singletons by ignoring the non-positioned nucleosomes.

Reply: The answer to this point probably depends on what one calls an array. Here we mean a regular spaced number of positioned nucleosomes. On the nucleosome centred frequency distribution (Fig 1D), such an array would appear as a series of regular peaks. This is certainly not the case, although in fact there are some minor peaks flanking the major nucleosome peak present on plots from all three data sets. These correspond to the approximately 5-10% of nucleosome that are present within pairs and larger arrays. It should be noted that these peaks are quite broad, as in fact their positions do vary to some degree. To accommodate, the possibility of missing arrays due to variable spacing, we have also plotted the data in Fig 1E in an alternative way. Here we show the number of nucleosomes lying within 200 bp of the first positioned nucleosome, and then the second and so on. This would detect irregular nucleosome arrays where the spacing varies considerably between each nucleosome. However again we find that only a small number of nucleosomes are present as pairs or larger arrays.

As the reviewer raises the possibility that we may still be missing arrays, where the space between positioned nucleosomes (known as the linker length) is greater than 50 bp, we have now included Fig EV2, which shows the distribution of spacing between positioned nucleosomes. This shows a bimodal distribution, with a small group of positioned nucleosomes sitting approx. 50bp apart (Pairs and greater), but the large majority sitting > 500 bp from each other. The text (p7) has been revised to accommodate this additional information.

4. Comment: Figure 2C and 2D don't exist but cited in the text.

Reply: An error left over from the previous revision. Corrected

5. Comment: The authors used ATAC-seq to show that positioned nucleosomes are enriched within open chromatin, addressing one of the previous concerns. However, the authors have to make sure that the ATAC-seq DNA fragments considered for this analysis are smaller than nucleosomes. As it is well known that ATAC-seq also map nucleosome positions and not solely represent open chromatin (Buenrostro et al Nat Methods 2013). This is particularly important if the tagged DNA is not size-selected prior to sequencing as was done with the published data used for this study.

Reply: The reviewer may have been misled in this section. We have in fact used published coordinates for open chromatin regions, not ATAC-seq data itself, and asked how positioned

nucleosomes partition between open and closed chromatin. We agree that ATAC-seq can be used to map nucleosomes, although this would be a challenge in the large human genome, however it would also not be useful in this context as we would be mapping positioned nucleosomes against positioned nucleosomes. Hence, we used mapped open chromatin regions. Examining the text again, we realise that it may be misleading, and both the Results section (p8) and the Methods (p18) have been revised for clarity.

6. Comment: The authors have shown the distribution of positioned nucleosomes centred around TSS of selected loci in Figure 3A and 3B, and genome-wide figure 3C. The authors have also shown the difference between the total enrichment of positioned nucleosomes within 300bp of TSS with relation to gene activity (Figure 3D). I recommend that the authors also show the difference in the distribution of positioned nucleosomes in relation to TSS of active vs non-active genes (the same gene sets in figure Figure 3D).

Reply: We have included an additional EV Figure (Fig EV3) to show the structures around the TSS. As also shown in the numerical form in Fig 3D, there are no qualitative differences between active and inactive genes.

7. Comment: The authors investigated the positioned nucleosome arrays that are enriched around RE1 (n=871) and CTCF (n=9,516) binding sites, which are in total around 10,000 sites in iPSCs. Taken that about 65% of these sites are surrounded positioned nucleosomes. One should expect for each of the 6,500 of RE1 and CTCF binding sites, there are more than three positioned nucleosomes. However, the authors made a major point that less than 0.1% of the positioned nucleosomes are within positioned arrays (more than 3 nucleosomes), which should be around 48 sites (0.1% of the total 48,840 positioned nucleosomes in iPSCs fig 1E). This doesn't make sense. The authors need to clarify this discrepancy.

Reply: The reviewer highlights a difference between two different approaches to analysis. In the first, we use an in-house algorithm to identify peaks associated with highly positioned nucleosomes (we validate this against published data to confirm that it gives similar numbers of positioned nucleosomes as previously estimated). Using this method, individual CTCF sites are likely to be dispersed across all nucleosome arrays sizes, including triplets, pairs and possibly even singletons, depending on the prominence of positioned nucleosomes at each site. In contrast, the second approach aligns all CTCF sites to the CTCF motif to create a consensus pattern for each cell state, this is a commonly used methodology to capture the average pattern at all CTCF sites across the genome. Due to averaging nearly 10,000 sites it has higher sensitivity and higher signal-to-noise.

8. Comment: The nucleosomes around the CTCF sites occupied only in NPC cells and not iPSC shown in figure 5E are clearly different from those shown in Figure 5B. The peaks are sharper and less positioned. The authors have completely ignored this obvious difference and stated as if the nucleosome positioning show similar pattern in both cases to strengthen their conclusion on the uncoupling between CTCF presence and nucleosome positioning. This issue have to be addressed in the revision.

Reply: The CTCF sites are not in fact different in NPC, but the signal is affected by increased noise due to having fewer sites to use for the consensus pattern. This was made clear in our previous revision, both in the text and Figure legend, and in the letter of reply (see below).

“It also proved to be a challenge to identify CTCF sites bound only in NPC cells as they did not appear to possess the CTCF consensus site that works in the pluripotent state. However by cross comparison of our small fragment data with the Encode NPC ChIP-seq data, we were able to identify 2,522 CTCF sites uniquely bound to the CTCF complex in NPC. **The smaller data set possesses slightly higher background noise than the pluripotent set using 9,478 sites**, however clearly shows a nucleosome array at the bound sites in NPC, but importantly that the array pre-exists in the pluripotent state prior to CTCF binding. It also shows a trend for the change in nucleosome positioning as CTCF binds. “

To help the reader we now show a line fitted to the data in addition to the raw data points. This is included in Fig 5D&E.

Referee #2:

1. Comment: Only 32% of the positioned nucleosomes in pluripotent cells maintain positioning in NPCs, and also NPCs have 8x more positioned nucleosomes. It is not clear from the analysis/discussion where these new nucleosomes are located in the NPCs. Much of the discussion seems more focused on the small percentage of common, positioned nucleosomes.

Reply: We have included an additional Data Table (new Table EV2) that shows the number of positioned nucleosomes across all 15 chromatin states for both pluripotent and NPC cells. Although there is 8-fold more nucleosomes in NPC, there is no substantial change in the distribution pattern of nucleosomes. Any differences are minor and certainly do not suggest that the large increase in positioned nucleosomes are focussed on any specific chromatin states.

2. Comment: In small genome organisms, positioned nucleosomes along gene bodies are likely due to boundary effects - NDRs flank the small coding regions. This provides a simple explanation for the lack of positioned nucleosomes along gene bodies in metazoans.

Reply: Yes, we totally agree, and this is what is interesting about nucleosomes in mammalian and human cells, as the loci where they are positioned are likely to be key regulatory regions. Interestingly, nucleosomes in these cells can't simply arise through boundary effects, as positioning does not appear to occur at the binding sites of all transcription sites (see Fig EV4), and as we show seems to persist even in the absence of bound protein complexes. We also point out in the manuscript that although the proportion of positioned nucleosomes in human cells is very low, the total number is still greater than the whole yeast nucleosome complement.

Acceptance

2nd April 2019

I am very pleased to accept your manuscript for publication in the next available issue of EMBO reports. Thank you for your contribution to our journal.

Corresponding Author Name: Adrian J Harwood

Manuscript Number: EMBOR-2018-46960V2